# A Comparative Study of Oncolytic Vaccinia Viruses Harboring Different Marine Lectins in Breast Cancer Cells

**DOI:** 10.3390/md21020077

**Published:** 2023-01-23

**Authors:** Yanrong Zhou, Qianpeng Wang, Qi Ying, Xiaomei Zhang, Ting Ye, Kan Chen, Gongchu Li

**Affiliations:** College of Life Sciences and Medicine, Zhejiang Sci-Tech University, Hangzhou 310018, China

**Keywords:** oncolytic vaccinia virus, marine lectins, apoptosis, virus replication

## Abstract

Our previous studies demonstrated that arming vaccinia viruses with marine lectins enhanced the antitumor efficacy in several cancer cells. This study aims to compare the efficacy of oncolytic vaccinia viruses harboring Tachypleus tridentatus lectin (oncoVV-TTL), Aphrocallistes vastus lectin (oncoVV-AVL), white-spotted charr lectin (oncoVV-WCL), and Asterina pectinifera lectin (oncoVV-APL) in breast cancer cells (BC). These results indicated that oncoVV-AVL elicited the highest anti-tumor effect, followed by oncoVV-APL, while oncoVV-TTL and oncoVV-WCL had lower effects in BC. Further studies showed that apoptosis and replication may work together to enhance the cytotoxicity of oncoVV-lectins in a cell-type dependent manner. TTL/AVL/APL/WCL may mediate multiple pathways, including ERK, JNK, Hippo, and PI3K pathways, to promote oncoVV replication in MDA-MB-231 cells. In contrast, these pathways did not affect oncoVV-TTL/AVL/APL/WCL replication in MCF-7 cells, suggesting that the mechanisms of recombinant viruses in MCF-7 (ER^+^, PR^+^) and MDA-MB-231 (TNBC) cells were significantly different. Based on this study, we hypothesized that ER or PR may be responsible for the differences in promoting viral replication and inducing apoptosis between MCF-7 and MDA-MB-231 cells, but the specific mechanism needs to be further explored. In addition, small-molecule drugs targeting key cellular signaling pathways, including MAPK, PI3K/Akt, and Hippo, could be conjunction with oncoVV-AVL to promote breast cancer therapy, and key pathway factors in the JNK and PI3K pathways may be related to the efficacy of oncoVV-APL/TTL/WCL. This study provides a basis for applying oncolytic vaccinia virus in breast carcinoma.

## 1. Introduction

Globally, breast cancer is the most common malignancy in women and is the second leading cause of death after lung cancer. Despite improvements in early diagnosis and treatments, the therapeutic efficacy remains unsatisfactory [1]. It is a highly complex systemic disease with various tumor subtypes differing in risk factors, symptoms, response to treatment, and consequences. According to the expression level of certain biomarkers, including the estrogen receptor (ER), progesterone receptor (PR), and human epidermal growth factor receptor-2 (HER2), breast cancer is divided into five main subtypes: Luminal A (ER^+^, PR^+^), Luminal B/HER2-negative, Luminal B/HER2-positive, HER2-enriched, and triple-negative breast cancer (TNBC) (ER^−^, PR^−^, HER2^−^) [2,3]. In particular, TNBC is the most aggressive subtype with metastatic disease, rapid disease progression, and poor survival outcomes [4]. Traditional therapies have side effects including inefficient curative effects, cognitive impairments, tumor metastasis increase, and resistance to established therapies. Therefore, novel therapeutic strategies for this malignancy are critically needed.

Oncolytic vaccinia virus (oncoVV) represents a potential strategy for cancer therapy based on its appealing features, such as extensive safety as a live vaccine, and efficient delivery to metastatic tumors [5,6,7,8,9]. Moreover, oncoVV can also be used as a gene expression vector due to its large size and its own enzyme expression systems [10]. To further promote antitumor efficiency, the use of harboring genes has become a strategy to modify the oncolytic vaccinia virus. 

Lectins, as a class of specific glycosyl-binding glycoproteins, preferentially recognize and bind carbohydrate complexes [11,12]. They show great potential for application in breast cancer therapy. For example, the agglutinins Bauhinia purpurea and Wisteria floribunda caused significant concentration-dependent antiproliferative effects on MCF-7 through lactate dehydrogenase leakage, cell cycle arrest, and reactive oxygen species generation [13]. Sialic acid-binding lectin from bullfrog eggs exhibits an anti-tumor effect against breast cancer cells including TNBC [14]. Ibacus novemdentatus lectin isolated from Arthropoda slipper lobster elicited cytotoxic effects on breast cancer cells MCF-7 and T47D through glycoconjugate interaction [15]. 

Currently, the clinical effectiveness of oncolytic virus as monotherapy remains limited, and, thus, researchers have been exploring various combinations. As cancer cells have evolved to alter key signaling pathways for enhanced cell proliferation, cancer progression, and metastasis, these cellular and molecular changes offer targets for BC therapy. Therefore, key molecules in relevant signaling pathways for cancer cells or/and immune cells, such as Raf/MEK/ERK, PI3K-AKT-mTOR, JNK, and Hippo/YAP, or immune pathways currently have the potential to synergize with the oncolytic virus to improve therapeutic efficacy [16].

We have previously generated four strains of oncoVV (a TK-deleted Western Reserve strain) by inserting marine lectins, including Tachypleus tridentatus lectin (TTL) [17], Aphrocallistes vastus lectin (AVL) [18,19,20,21], white-spotted charr lectin (WCL) [22], and Asterina pectinifera lectin (APL). In the current study, we sought to advance our previous research by assessing the antitumor efficacy of four strains of vaccinia virus in breast cancer cells (BC). For proof-of-principle studies, two typical BC subtypes were chosen: the ER^+^ subtype, which is the most common subtype, and the triple-negative subtype, which is the most lethal subtype. The replication mechanisms of recombinant viruses in MCF-7 (ER^+^, PR^+^) and MDA-MB-231 (TNBC) were further compared. These observations may lay the foundation for the search for combination drugs of recombinant viruses and help the use of these oncolytic vaccinia viruses in gene therapy for breast cancer.

## 2. Results

### 2.1. Cytotoxicity of oncoVV-AVL/APL/TTL/WCL in BC 

To assess the cytotoxicity of recombinant viruses, an MTT assay was performed to determine cell viability. MCF-7 and MDA-MB-231 cells were infected with recombinant viruses at 2, 5, or 10 multiplicity of infection (MOI) for 24, 48, and 72 h. OncoVV served as the control. As shown in Figure 1, exogenous lectins expression enhanced the cytotoxicity of oncoVV in BC. Figure 1a showed that oncoVV-TTL/AVL/APL/WCL led to the cell viability dropping to 41%, 22%, 29%, and 62%, respectively, indicating that the effects of recombinant viruses on MCF-7 cells were oncoVV-AVL > oncoVV-APL > oncoVV-TTL > oncoVV-WCL. Furthermore, AVL showed obvious cytotoxicity to MCF-7 cells with 2 MOI at 24 h post infection, whereas the others had no effect on BC. Figure 1b shows that the effects of recombinant viruses on MDA-MB-231 cells were oncoVV-AVL (23%) > oncoVV-APL (45%) > oncoVV-TTL (76%) ≈ oncoVV-WCL (77%). These results indicated oncoVV-AVL elicited the highest anti-tumor effect, followed by oncoVV-APL, while oncoVV-TTL and oncoVV-WCL had lower effects on BC.

### 2.2. Apoptotic Effect of oncoVV-lectins in BC

Research increasingly shows that lectins have remarkable abilities in inducing apoptosis [23,24]. To study the inducing apoptosis effect of recombinant viruses on BC, flow cytometry was adopted to examine apoptotic rate. As shown in Figure 2a, compared with PBS and oncoVV controls, oncoVV-AVL/APL/WCL enhanced apoptosis in MCF-7 cells. The proportions of apoptotic cells were 22%, 17%, and 14% for in oncoVV-AVL, oncoVV-APL, and oncoVV-WCL, correspondingly (Figure 2c), indicating that the effects of lectins enhancing apoptosis were oncoVV-AVL > oncoVV-APL > oncoVV-WCL. Figure 2b,d show that compared with controls, oncoVV-lectins induced significantly higher levels of apoptosis, 19% (oncoVV-TTL), 72% (oncoVV-AVL), 22% (oncoVV-APL), and 6% (oncoVV-WCL), respectively. Flow cytometry showed that the highest apoptosis was induced by oncoVV-AVL in both MCF-7 and MDA-MB-231 cells, but MDA-MB-231 cells infected with oncoVV-AVL induced much higher apoptosis.

### 2.3. Lectins Harboring Improved the Replication of Oncolytic Vaccinia Viruses in BC

To study the underlying mechanisms of the antiproliferative effects of oncoVV-lectins, the replication was determined by TCID_50_ in BC at the time points of 0, 12, 24, 36, and 48 h, respectively. Figure 3a shows that the peak reproductive number of oncoVV was observed at 36 h, which was increased by approximately 5.3 times. Compared to the control, the replication of oncoVV-TTL/AVL/APL/WCL was increased by approximately 4-, 84-, 56-, and 11-fold, respectively, within 36 hours, suggesting that the viral copies were oncoVV-AVL > oncoVV-APL > oncoVV-TTL > oncoVV-WCL in MCF-7 cells. The results of A27L [25,26,27] expression were consistent with the above results (Figure 3b). As shown in Figure 3c, the amount of viruses increased by about 2.4-fold in 48 hours compared to the starting point. In addition, the replication of oncoVV-TTL/AVL/APL/WCL was 7-, 242-, 122-, and 31-fold higher than that of oncoVV control at 36 h, indicating that oncoVV-AVL showed the highest replication, followed by oncoVV-APL, while oncoVV-TTL/WCL had the lower replication. Moreover, the A27L expression had a strong positive correlation with the virus yields (Figure 3d). In short, these results demonstrated that arming oncolytic vaccinia viruses with marine lectins promoted viral replication in a cell-type dependent manner. 

### 2.4. Pathways Associated with the Replication Ability of OncoVV-lectins

The Raf/MEK/extracellular signal-regulated kinase (ERK) signaling pathway plays a crucial role in almost all cell functions. It is reported that lectins can result in ERK molecular alterations [28], which is required for viral replication in some tumor cells [18,20,29]. Our results suggested that Sorafenib (Raf inhibitor) or U0126 (MEK inhibitor) significantly inhibited the replication of oncoVV-APL but not the others in MCF-7 cells (Figure 4a). Figure 4c showed that oncoVV reduced the phosphorylation of ERK compared with PBS, but the oncoVV-AVL/APL treatments restored the expression of phosphorylation of ERK, indicating that APL enhanced oncoVV replication depending on Raf/MEK/ERK signal pathway while oncoVV-AVL replication may not depend on this pathway in MCF-7 cells. As seen in Figure 4b, Sorafenib or U0126 did not affect the replication of oncoVV or oncoVV-TTL, but significantly restrained the replication of oncoVV-AVL/APL/WCL. Furthermore, Western blot results showed that oncoVV-TTL/AVL treatments upregulated the phosphorylation of ERK but not the others (Figure 4d), suggesting that oncoVV-TTL/AVL could promote the phosphorylation of ERK in MDA-MB-231 cells. However, the functions of ERK activation might be different between oncoVV-AVL and oncoVV-TTL. These findings showed that AVL/APL may improve oncoVV replication by activating Raf/MEK/ERK pathway in a cell-type dependent manner.

DNA virus can utilize the JNK pathway to establish a regulatory circuit to promote replication [30]. As shown in Figure 5a, SP600125, a JNK inhibitor, slightly promoted the replication of oncoVV, and had no effect on recombinant viruses replication, suggesting that JNK activity did not affect the replication of vaccinia virus. Figure 5b showed that the replication of recombinant viruses was dramatically reduced by SP600125, which did not alter the replication of oncoVV, indicating that lectins enhanced the replication of oncoVV may depend on JNK activities in MDA-MB-231 cells. These results indicated that the JNK pathway had utterly opposite effects on the replication of recombinant viruses in MCF-7 and MDA-MB-231 cells, pending further investigation.

Recent studies show that the Hippo pathway inhibits innate immunity to enhance viral replication [31]. In order to evaluate the effect of the pathway on the replication of oncoVV harboring different lectins, MST1/2 inhibitor XMU-MP-1 was used. In our study, we found that XMU-MP-1 decreased the replication of both oncoVV and oncoVV-lectins (Figure 6a). It was reported that the protein level of YAP was negatively correlated with the expression of ER/PR [32], and no obvious YAP bands were detected in MCF-7 cells, indicating that the higher replication caused by recombinant viruses may be independent of MST1/2 activities in MCF-7 cells. However, as seen in Figure 6b, XMU-MP-1 had no effect on the replication of oncoVV and oncoVV-TTL/WCL, but significantly decreased oncoVV-AVL/APL replication in MDA-MB-231 cells. It is well-known that the MST1/2-LATS cascade can negatively regulate the activation of YAP [33]. Consistently, lower expression of YAP had been detected under the oncoVV-AVL/APL treatments (Figure 6c), suggesting that AVL/APL enhanced replication by regulating the Hippo/YAP pathway in MDA-MB-231 cells. Additionally, the replication of oncoVV-WCL was not affected by XMU-MP-1 (Figure 6b), whereas YAP degradation was increased by oncoVV-WCL (Figure 6c), suggesting that oncoVV-WCL replication in MDA-MB-231 cells may not depend on YAP activity. Taken together, these results showed that oncoVV-AVL/APL replication in MDA-MB-231 cells depended on Hippo/YAP pathway.

The PI3K/Akt pathway is involved in the assembly and budding of poxviruses [34]. In order to evaluate the effect of the PI3K on the replication of recombinant viruses, the PI3K inhibitor KY12420 was used. Figure 7a shows that KY12420 had no effect on viral replication, suggesting that the replication of both oncoVV and oncoVV-lectins were independent of PI3K pathway in MCF-7 cells. By contrast, the replication of oncoVV-TTL/AVL/WCL was significantly decreased in the presence of KY12420 (Figure 7b), which had no effect on the replication of oncoVV and oncoVV-APL, indicating that PI3K may be involved in the replication of oncoVV-TTL/AVL/WCL in MDA-MB-231 cells. We inferred that TTL/AVL/WCL enhanced oncoVV replication in MDA-MB-231 cells by interfering with PI3K signaling pathway. 

## 3. Discussion

It is possible for oncolytic viruses to treat refractory breast cancer owing to direct tumor cell lysis by virtue of their selective replication in cancer cells [35,36,37,38]. Lectins selectively bind to specific sugar sequences on the cell surface, which are exploited to exclusively bind cancer cells and exert antitumor activities. Previously, we used vaccinia viruses as delivery vectors to express marine lectin genes, which had shown excellent anti-tumor activity. In this study, we compared the antitumor actions of four recombinant viruses and further explored their mechanisms. The results showed that lectins enhanced the anti-tumor effects by inducing apoptosis and promoting replication. Among them, oncoVV-AVL had the best anti-tumor effect with the highest apoptosis and replication.

ER may be responsible for the differences in inducing apoptosis between MCF-7 and MDA-MB-231 cells. Song RX et al. found that ER-dependent activation of ERK inhibited apoptosis through ERα interacts with SHC, Src, and Ras [39]. It was found that ER mediated JNK or PI3K/AKT pathways to inhibit cell apoptosis [40,41]. A recent study found that YAP had the opposite effect in ER-positive and triple-negative breast cancer. In most malignant tumors, gene amplification and overexpression of YAP were shown to promote tumor progression. However, in ER-positive breast cancer, YAP protein was less expressed than normal breast tissue, and the level of YAP expression tended to be favorable for prognosis [32]. In contrast, recombinant viruses may induce higher apoptosis in absence of ER in MDA-MB-231 cells. Based on the above studies, we supposed that ER inhibited MCF-7 apoptosis induced by lectins through regulating signal pathways or antiviral immunity, but the specific mechanism needs to be further explored.

The replication mechanisms of recombinant viruses were in a cell type-dependent manner, and lectins have different effects in promoting viral replication. In our study, APL enhanced oncoVV replication depending on the Raf/MEK/ERK signal pathway in MCF-7 cells, but the exact opposite was observed in MDA-MB-231 cells. JNK activity did not affect the vaccinia virus replication in MCF-7 cells, while JNK inhibitors inhibited the replication of all recombinant viruses in MDA-MB-231 cells. In addition, oncoVV-AVL/APL replication in MDA-MB-231 cells depended on the Hippo/YAP pathway, while the opposite phenomenon was observed in MCF-7 cells. The PI3K inhibitor had no effect on all viruses replication in MCF-7 cells, but PI3K could significantly manipulate the replication of oncoVV-TTL/AVL/WCL in MDA-MB-231 cells. Many studies have shown that ER and PR can affect viral replication and antiviral response. For example, ER and PR can inhibit innate and acquired immunity, thus changing susceptibility to HIV [42]. ER have been found to inhibit Zika viruses infection and the replication process [43]. Similarly, our previous study found that AVL could repress the antiviral response in Huh7 cells to facilitate viral replication [20]. In summary, the replication mechanisms of recombinant viruses between MCF-7 and MDA-MB-231 cells were completely different. ER or PR may affect viral replication and antiviral immunity by changing the lectin-signaling pathways interaction in BC, and lectins have different effects in promoting viral replication, which may be related to ligand (lectins)-receptor interactions. However, the specific mechanisms need to be further explored.

Key molecules in relevant signaling pathways had the potential to synergize with oncoVV-lectins to improve therapeutic efficacy. In MDA-MB-231 cells, we found that Sorafenib or U0126 significantly restrained oncoVV-AVL replication, and oncoVV-AVL infection triggered activation of cellular phosphorylation of ERK, suggesting that AVL improved viral replication by activating ERK pathway. Furthermore, the replication of oncoVV-AVL was dramatically reduced by SP600125 and KY12420, suggesting that JNK and PI3K could affect the replication of oncoVV-AVL as well. In addition, XMU-MP-1 significantly decreased the replication of oncoVV-AVL, and lower expression of YAP had been detected in MDA-MB-231 cells treated with oncoVV-AVL, suggesting that AVL enhanced replication by regulating the Hippo/YAP pathway. Based on these studies, we inferred that small-molecule drugs targeting key cellular signaling pathways, including MAPK, PI3K/Akt, and Hippo, could serve as conjunctions with oncoVV-AVL to promote breast cancer therapy. In addition, key pathway factors in the JNK and PI3K pathways may be related to the efficacy of oncoVV-APL/TTL/WCL. However, the replication mechanism of recombinant viruses in MCF-7 cells is still poorly understood, and needed to be further explored.

## 4. Materials and Methods

### 4.1. Cell Lines and Cell Culture

Human embryonic kidney cell line HEK 293A and human breast carcinoma cell lines MCF-7 and MDA-MB-231 were obtained from the Chinese Academy of Sciences. Cell lines were cultured in DMEM (Gibco, Thermo Fisher Scientific, Waltham, MA, USA) in combination with 10% fetal bovine serum (FBS) (Hyclone Laboratories, Dunedin, Otago, New Zealand) and 1% penicillin-streptomycin.

### 4.2. Virus Amplification and Crude Purification 

Cells were infected with viruses at a titer of 0.01 MOI. After the cells were completely diseased (about 48 h), samples were collected and freeze-thawed for 3 times at −80℃ to release the viruses. The collected samples were centrifuged at a low speed of 2000 rpm/30 min to remove cell debris. Then, the centrifugation was performed at a high speed at 12,000 rpm/30 min to collect virus precipitates. In addition, the virus precipitates were re-suspended with PBS, and the centrifugation was repeated three times. Finally, the precipitates re-suspended with PBS and stored in the refrigerator at −80℃.

### 4.3. Cell Viability Assay 

Cell viability was assessed by MTT as described before [15,16,17,18]. Cells (8×10^3^ cells/well) were treated with recombinant viruses at 2, 5 or 10 MOI for 24, 48 and 72 h post infection. Then the absorbance at 490 nm was determined by microplate reader (Multiskan, Thermo Scientific, Waltham, MA, USA).

### 4.4. Cell Apoptosis Detection

Cells (4 × 10^5^ cells/well) were seeded into plates overnight, and then 5 MOI oncoVV-TTL, oncoVV-AVL, oncoVV-APL, or oncoVV-WCL, respectively, were added. PBS and oncoVV served as controls. The above treated cells were collected and stained with Annexin V-FITC and propidium iodide (PI) (BD Biosciences, San Jose, CA, USA). Flow cytometry was used to analyze the samples (AccuriC6, BD Biosciences, San Jose, CA, USA).

### 4.5. Detection of Viral Replication Ability

The viral replication abilities of oncoVV and oncoVV-lectins in MCF-7 and MDA-MB-231 cells were detected by a TCID_50_ assay. HEK 293A cells were plated in 24-well plates (8 × 10^4^ cells/well) for 12 h before being infected with different recombinant viruses (5 MOI/well). Samples were collected at 0, 12, 24, 36, 48 h. To release virus particles from cells, the collected samples were frozen and thawed repeatedly 3 times between −80 °C and room temperature, and then each well’s virus suspension samples were collected for titer determination. Finally, the virus concentration was successively diluted with DMEM containing 2% FBS (10^−1^~10^−9^). The diluted virus was then added to 96-well plates of HEK 293A cells (4 × 10^3^ cells/well) with 8 replicates per concentration. After 7 days of culture, the number of viral plaque holes was counted, and the virus titer was 7 × 10^(d + 0.5)^ PFU/mL (d = sum of positive proportion) [44].

To explore how TTL, AVL, APL, and WCL affected oncoVV replication, different agents (Selleck Chemicals LLC, Houston, TX, USA) were added after cells were maintained for 12 h, and the viruses were added to infect the cells about 1 h later. Cell samples were collected at 48 h post infection. The experimental steps for testing the viral titers were performed as described above. Agents used in MCF-7 were Sorafenib (3 μmol/L), U0126 (15 μmol/L), SP600125 (3 μmol/L), KY12420 (4.5 μmol/L), and XMU-MP-1 (10 μmol/L). The agents used in MDA-MB-231 cells were Sorafenib (3 μmol/L), U0126 (15 μmol/L), SP600125 (3 μmol/L), KY12420 (4.5 μmol/L), and XMU-MP-1 (0.2 μmol/L). 

### 4.6. Western Blot Analysis

Cells infected with viruses for 36 h at 5 MOI were collected and lysed. Cytosolic proteins samples were quantified, then separated by 15% SDS-PAGE and transferred to PVDF membrane (Millipore, Bedford, MA, USA). The membrane was blocked with 5% non-fat milk at room temperature and then incubated with primary antibodies (1:1000 dilution) at 4 ℃ overnight. After washing, the blots were incubated with secondary antibodies (1:5000 dilution) for 1 h at room temperature. Finally, the protein bands on the membrane were scanned with a chemiluminescence image system. The flowing primary antibodies GAPDH, ERK, p-ERK and YAP were purchased from CST (Cell Signaling Technology, Boston, MA, USA), while A27L was obtained from Abcam (Abcam Plc, Cambs, UK). The corresponding secondary antibodies HRP-conjugated Goat anti-Rabbit IgG and HRP-conjugated Goat anti-Mouse IgG were purchased from ABclonal (ABclonal Technology, Wuhan, Hubei, China).

### 4.7. Data Analysis

GraphPad Prism 8.0 software (GraphPad Software, San Diego, USA) was used for analyses. The one-way ANOVA test was performed for comparison among groups. All results were shown in mean ± SEM, and p < 0.05 or p < 0.01 was considered statistically significant.

## 5. Conclusions

The present study demonstrated that marine lectins TTL/AVL/APL/WCL significantly enhanced the cytotoxicity of oncoVV in BC by inducing apoptosis and promoting replication. These results indicated oncoVV-AVL elicited the highest anti-tumor effect, followed by oncoVV-APL, while oncoVV-TTL and oncoVV-WCL had lower anti-tumor effects in both MCF-7 and MDA-MB-231 cells. Further studies showed that apoptosis and replication may work together to enhance the cytotoxicity of oncoVV-lectins in a cell-type dependent manner. TTL/AVL/WCL may mediate multiple pathways, including ERK, JNK, Hippo, and PI3K pathways, to promote oncoVV replication in MDA-MB-231 cells. In contrast, these pathways did not affect oncoVV-TTL/AVL/WCL replication in MCF-7 cells, the mechanisms of recombinant viruses in MCF-7 and MDA-MB-231 cells may be significantly different. In addition, we inferred that small-molecule drugs targeting key cellular signaling pathways, including MAPK, PI3K/Akt, and Hippo, could be conjunction with oncoVV-AVL to promote breast cancer therapy, and key pathway factors in the JNK and PI3K pathways may be related to the efficacy of oncoVV-APL/TTL/WCL. This study provides a basis for applying oncolytic vaccinia virus in breast carcinoma. 

## Figures and Tables

**Figure 1 marinedrugs-21-00077-f001:**
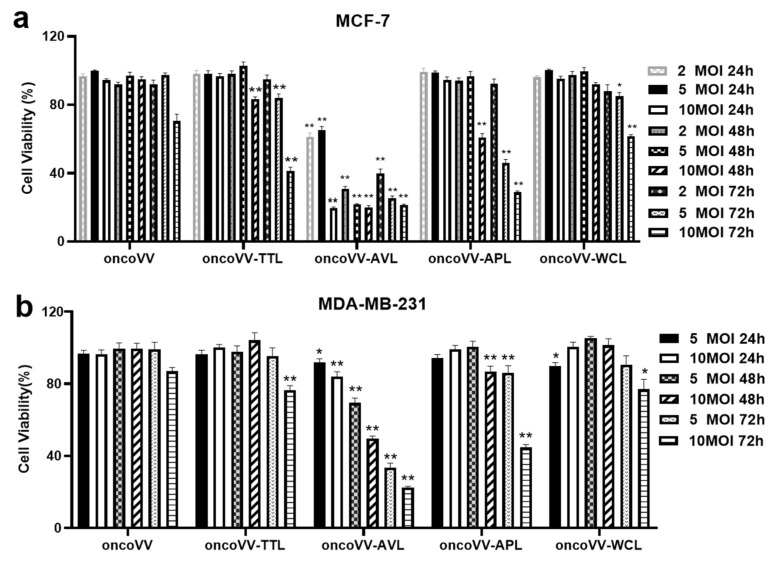
OncoVV-lectins inhibited cell viability in MCF-7 (**a**) and MDA-MB-231 (**b**) cells. Cells were infected with oncoVV-TTL, oncoVV-AVL, oncoVV-APL, or oncoVV-WCL, respectively, at 2, 5, or 10 MOI for 24, 48, and 72 h. OncoVV was used as the control. Data represented the mean ± SEM from at least three independent experiments (* *p* < 0.05, ** *p* < 0.01).

**Figure 2 marinedrugs-21-00077-f002:**
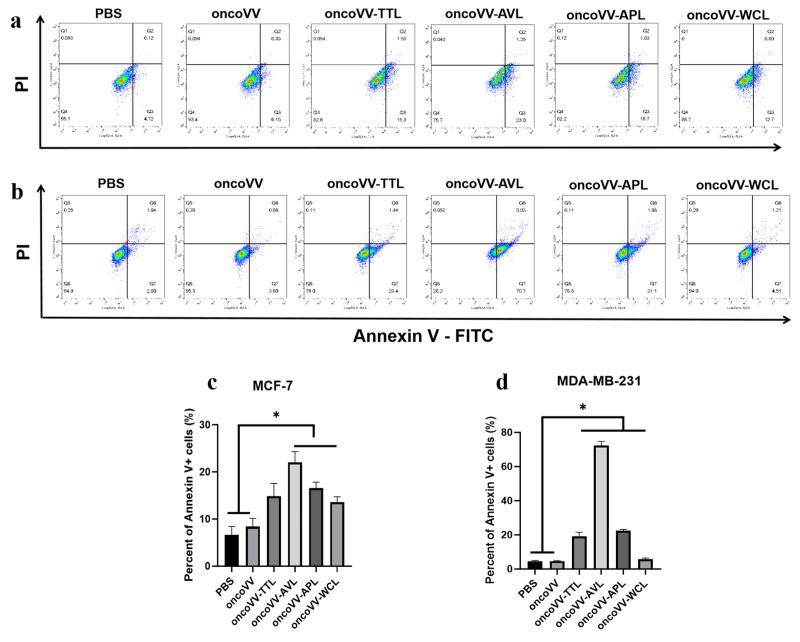
OncoVV-lectins inducing apoptosis in MCF-7 (**a**) and MDA-MB-231 (**b**) cells. Cells were treated with oncoVV-TTL, oncoVV-AVL, oncoVV-APL, or oncoVV-WCL, respectively, at 5 MOI for 36 h. PBS or oncoVV served as the controls. (**c**,**d**) The percentage of apoptotic cells. Data represent the mean ± SEM from at least three independent experiments (* *p* < 0.05).

**Figure 3 marinedrugs-21-00077-f003:**
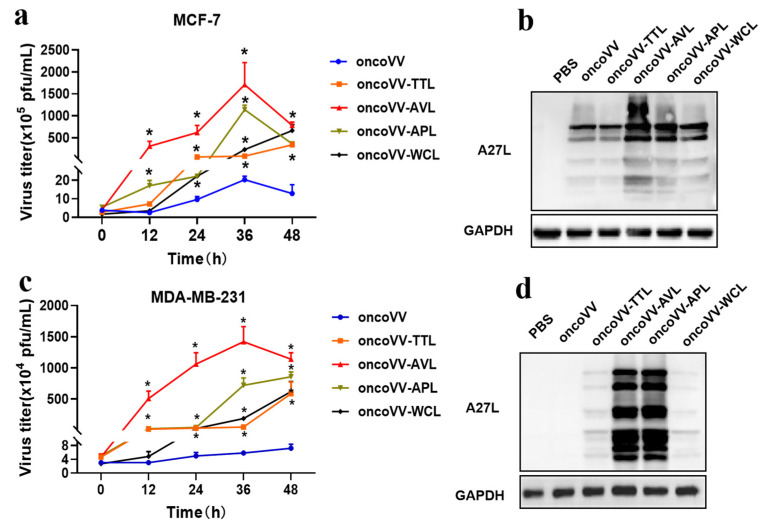
OncoVV-lectins replication in MCF-7 (**a**) and MDA-MB-231 (**c**) cells. OncoVV or recombinant viruses were used at the concentration of 5 MOI, and viral replication was determined by TCID_50_ assay in HEK293A cells. Values represent the mean ± SEM from three biological replicates (* *p* < 0.05). The expression levels of A27L were detected by Western blot in MCF-7 (**b**) and MDA-MB-231 (**d**) cells. PBS and oncoVV served as controls, and GAPDH was a loading control.

**Figure 4 marinedrugs-21-00077-f004:**
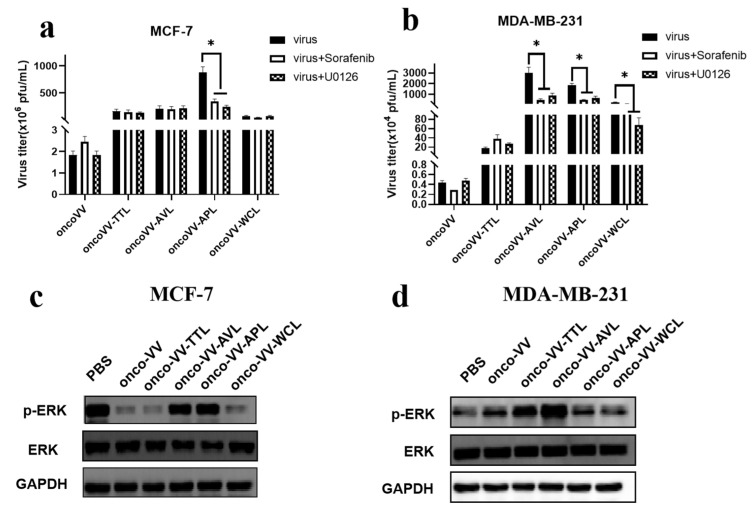
The effects of the Raf/MEK/ERK pathway on the replication of oncoVV-lectins. MCF-7 (**a**) and MDA-MB-231 (**b**) cells were infected with oncoVV-TTL, oncoVV-AVL, oncoVV-APL, or oncoVV-WCL in the presence of Sorafenib or U0126 at 5 MOI for 36 h (* *p* < 0.05). YAP expressions were detected by Western blot in MCF-7 (**c**) and MDA-MB-231 (**d**) cells. OncoVV served as the control, and GAPDH was the loading control.

**Figure 5 marinedrugs-21-00077-f005:**
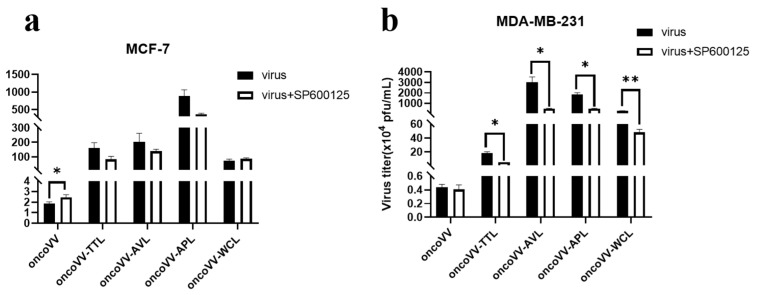
Virus yields in the presence of JNK inhibitor. MCF-7 (**a**) and MDA-MB-231 (**b**) cells were infected with oncoVV-TTL, oncoVV-AVL, oncoVV-APL, or oncoVV-WCL in combination with SP600125 at 5 MOI for 36 h (* *p* < 0.05, ** *p* < 0.01). OncoVV served as the control.

**Figure 6 marinedrugs-21-00077-f006:**
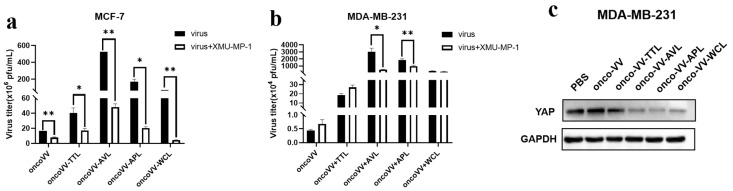
The interaction of exogenous lectins with the Hippo/YAP pathway. The virus titers in MCF-7 (**a**) and MDA-MB-231 (**b**) cells with the presence of XMU-MP-1 (* *p* < 0.05, ** *p* < 0.01). (**c**) Expressions of YAP were determined by Western blotting. PBS or oncoVV served as the controls, and GAPDH was a loading control.

**Figure 7 marinedrugs-21-00077-f007:**
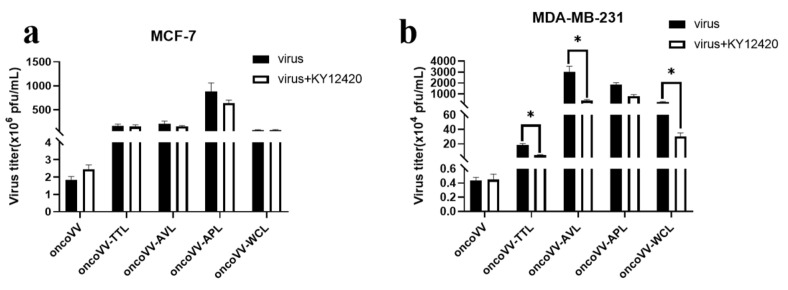
Virus yields in the presence of PI3K inhibitor. MCF-7 (**a**) and MDA-MB-231 (**b**) cells were infected with oncoVV-TTL, oncoVV-AVL, oncoVV-APL, or oncoVV-WCL in the presence of KY12420 at 5 MOI for 36 h (* *p* < 0.05). OncoVV served as control.

## Data Availability

Data are contained within the article.

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
