# Peer review of "A Comparative Study of Oncolytic Vaccinia Viruses Harboring Different Marine Lectins in Breast Cancer Cells"

_marinedrugs, 2023, doi:10.3390/md21020077_

Round 1

Reviewer 1 Report

1. Page 2: The following sentence should be checked. "They hold not only potential for cancer diagnosis and prognosis, but also show great potential for application in breast cancer therapy."

2. Page 2, Section 2.1: Write the full form of 'MOI' when appears for the first time.

3. Page 2, Figure 1a: Please change the pattern of bars indicating '2 MOI 24h' and '2 MOI 24h'. At this moment, tough to differentiate.

4. Page 3, Figure 2a: Please provide a magnified view. Replace the phrase 'Statistical results of apoptosis rate' with more meaningful one.

5. Page 4: "suggesting that the replication of oncoVV-AVL partly depended on Raf/MEK/ERK pathway while oncoVV-TTL/APL/WCL replication may be independent of this pathway in MDA-MB-231cells." But the expression of oncoVV-TTL is much greater than oncoVV-APL or oncoVV-WCL in Figure 4d. Please check that statement. 

6. The discussion was not up to the mark. Authors focused too much on the pathways. They suggested to modify it adding a few more references. 

7. Authors summerized that, replication mechanisms of recombinant viruses were completely different in MCF-7 and MDA-MB-231 cell lines. But how do these cell types differ from each other? Why those pathways did not work for MCF-7 cells? Authors are suggested to write a few sentences focusing on that point. Even a speculation will be interesting.

8. Please correct small grammatical mistakes. Example: "....had a very slightly impact".

Author Response

Point 1: Page 2: The following sentence should be checked. "They hold not only potential for cancer, but also show great potential for application in breast cancer therapy.

Response 1: Thanks for kindly reminder. The sentence was revised as "They show great potential for application in breast cancer therapy".

Point 2: Page 2, Section 2.1: Write the full form of 'MOI' when appears for the first time.

Response 2: We have written the full form of 'MOI' when appears for the first time, and highlighted by red color. (Results section 2.1)

Point 3: Page 2, Figure 1a: Please change the pattern of bars indicating '2 MOI 24h' and '2 MOI 24h'. At this moment, tough to differentiate.

Response 3: Thanks for the comments. We have changed the pattern of bars indicating '2 MOI 24h'.

Point 4: Page 3, Figure 2a: Please provide a magnified view. Replace the phrase 'Statistical results of apoptosis rate' with more meaningful one.

Response 4: Thanks for the suggestions. We have uploaded a magnified Figure 2, and the phrase 'Statistical results of apoptosis rate' was replaced as 'The percentage of apoptotic cells', and highlighted by red color.

Point 5: Page 4: "suggesting that the replication of oncoVV-AVL partly depended on Raf/MEK/ERK pathway while oncoVV-TTL/APL/WCL replication may be independent of this pathway in MDA-MB-231cells." But the expression of oncoVV-TTL is much greater than oncoVV-APL or oncoVV-WCL in Figure 4d. Please check that statement. 

Response 5: Thank you for pointing this out. The statement was corrected as "suggesting that oncoVV-TTL/AVL could promote the phosphorylation of ERK in MDA-MB-231 cells. However, the functions of ERK activation might be different between oncoVV-AVL and oncoVV-TTL" in the manuscript, and highlighted by red color.

Point 6: The discussion was not up to the mark. Authors focused too much on the pathways. They suggested to modify it adding a few more references. 

Response 6: Thanks for the suggestions. Some references and content have been added to the discussion section, which was highlighted with red color.

Point 7: Authors summerized that, replication mechanisms of recombinant viruses were completely different in MCF-7 and MDA-MB-231 cell lines. But how do these cell types differ from each other? Why those pathways did not work for MCF-7 cells? Authors are suggested to write a few sentences focusing on that point. Even a speculation will be interesting.

Response 7: Thanks for the suggestions. We have rewritten the discussion as your suggestion, and highlighted by red color.

Point 8: Please correct small grammatical mistakes. Example: "....had a very slightly impact".

Response 8: We are sorry for these grammatical mistakes. We revised the descriptions.

Reviewer 2 Report

This is a comparison study of previous characterized 4 marine-lectins-armed oncolytic viruses (VV) on viral replication, apoptosis and dependence of certain signaling pathways. All work was performed in vitro, and no tumor models have been utilized.

Data are presented to show that some lectins affect the replication of oncolytic vaccinia virus, and it potential in induction of apoptosis in infected human breast cancer cells. In this case, AVL and APL are more potent than others in both MCF7 and MDA-MB-231 cancer cells.

Main strengths and weakness:

1.      The collected data are interesting, yet the authors could improve the quality of the study by strengthen the Introduction, and Discussion, and provide some hypothesis if possible.

2.      Figure 1. It is shocking to see that the oncoVV (tk-deleted VV, WR strain) induced little, if any, cytotoxicity even at MOI of 10 for 72 h in both MCF7 and MDA-MB-231 cells. This reviewer is quite familiar with this type of oncolytic virus and the breast cancer cell lines used and expected to see significant cytotoxicity induced under this set of conditions. Could authors define the MOI differently from others? Looking at the previous papers published by the authors, it seemed that authors might have used crude viruses without purification, as no procedure of virus purification has been described.

3.      The authors have done a great job in looking into the relationship between the signaling pathway versus viral replication of vaccinia virus. Unfortunately, the results in MCF-7 versus MDA-MB-231 are quite different in many cases, the authors did not provide any explanations or some hypothesis. As such, this study provides little value to others.

4.      There are a variety of death pathways, such as apoptosis, necroptosis, ferroptosis, etc. The literature indicated that vaccinia virus induces cell death mainly through two pathway, apoptosis and necroptosis.

5.      The authors have claimed that these lectins have different effects in promoting viral replication and inducing apoptosis of host cells. Could this be due to a simple possibility that their activities are related to the levels of the receptors expressed on the cancer cells, leading to activation of downstream signaling cascade that activate/suppress innate immunity that affect viral replication?  As the authors have analyzed the downstream key signaling pathways, but it would be nice to know that ligand (lectin)-receptor interactions in the first place.

Minor points:

1.      Introduction. When introducing the topic of oncolytic vaccinia virus (poxvirus), the authors cited reviews published in 2009; 2011 and 2015. In fact, there are much more recent comprehensive and authoritative reviews on the topic (published in 2019) that provide more up-to-date authoritative reviews and more suitable for citations for this purpose: (1). PMID: 30626434. (2). PMID: 30919708. Unlike some cited older papers, the authors of these recent review articles have been and continue working in the field for over 20 years and going and their reviews are more authoritative and up-to-date.

2.      Methodology: Descriptions of some methodologies are not detailed enough. For example, when authors have used inhibitors (such as sorafenib) together with VV, when was the inhibitor added, and how often (just once)? A lot of detailed information need to be added.

3.      Fig. 3 a,b. Based on the current plots, it seems that oncoVV does not replicate at all. This is very misleading. In addition, the description in the Results is also poor as there are no information on the basal level of replication (or accumulated virus) from oncoVV.

4.      Figs 3-7. The authors stated that they conducted TCID50 assay to measure quantities of infectious virus, yet the Y-axis indicates pfu/ml. How did the authors convert to pfu? (It has not been mentioned at all). This should be described in Methods.

5.      Figs. 4-5. At which time point were the virus titers determined?

6.      Fig. 6c. Lanes are not labeled.  In addition, there should be 12 lanes, instead of 6 lanes.

7.      Discussion: Authors have discussed the confusing data collected from these two breast cancer cell lines (MCF7 and MDA-MB-231). However, one thing is clear that data drawn by the authors supported that statement that oncolytic virus combined with some specific inhibitor or activator of specific signaling pathway may work synergistically, leading to improved therapy. In this regard, there is a review to support your discussion (PMID: 36221123. Improving cancer immunotherapy by rationally combining oncolytic virus with modulators targeting key signaling pathways. Mol Cancer. 2022;21:196.).

8.      The following existing references need correct or additional information:

Ref #1. Ferlay J. et al. The correct page numbers are, 778-789.

Ref #18. Authors’ own paper. The correct volume number is 20. not 11.

Ref #25. Jiang Q. et al. The missing page numbers are, 17-28.

Author Response

Response to Reviewer 2 Comments

Point 1: The collected data are interesting, yet the authors could improve the quality of the study by strengthen the Introduction, and Discussion, and provide some hypothesis if possible.

Response 1: Thank you for your constructive comments. The introduction and discussion were rewritten based on your suggestions, and highlighted by red color.

Point 2: Figure 1. It is shocking to see that the oncoVV (tk-deleted VV, WR strain) induced little, if any, cytotoxicity even at MOI of 10 for 72 h in both MCF7 and MDA-MB-231 cells. This reviewer is quite familiar with this type of oncolytic virus and the breast cancer cell lines used and expected to see significant cytotoxicity induced under this set of conditions. Could authors define the MOI differently from others? Looking at the previous papers published by the authors, it seemed that authors might have used crude viruses without purification, as no procedure of virus purification has been described.

Response 2: Thank you for your constructive comments. Crude viruses were used in the study, and the method of virus amplification and crude purification was added in Materials and Methods section, and highlighted by red color.

Point 3: The authors have done a great job in looking into the relationship between the signaling pathway versus viral replication of vaccinia virus. Unfortunately, the results in MCF-7 versus MDA-MB-231 are quite different in many cases, the authors did not provide any explanations or some hypothesis. As such, this study provides little value to others.

Response 3: Thanks for the suggestions. We provide some explanations or hypotheses for the different replication mechanisms of recombinant viruses between MCF-7 and MDA-MB-231 cells in the discussion, and highlighted by red color in discussion section.

Point 4: There are a variety of death pathways, such as apoptosis, necroptosis, ferroptosis, etc. The literature indicated that vaccinia virus induces cell death mainly through two pathway, apoptosis and necroptosis.

Response 4: Thank you for your constructive comments. We do plan to further explore more death pathways induced by oncolytic vaccinia viruses in the near future, but this study focuses on the death pathways of apoptosis and necroptosis.

Point 5: The authors have claimed that these lectins have different effects in promoting viral replication and inducing apoptosis of host cells. Could this be due to a simple possibility that their activities are related to the levels of the receptors expressed on the cancer cells, leading to activation of downstream signaling cascade that activate/suppress innate immunity that affect viral replication? As the authors have analyzed the downstream key signaling pathways, but it would be nice to know that ligand (lectin)-receptor interactions in the first place.

Response 5: Thank you for your constructive comments. We have rewritten the discussion as your suggestion, and highlighted by red color. The details are as follows:

‘ER may be responsible for the differences in inducing apoptosis between MCF-7 and MDA-MB-231 cells. Song RX et al. found that ER-dependent activation of ERK inhibited apoptosis through ERα interacts with SHC, Src, and Ras [39]. It was found that ER mediated JNK or PI3K/AKT pathways to inhibit cell apoptosis [40, 41]. A recent study found that YAP had the opposite effect in ER-positive and triple-negative breast cancer. In most malignant tumors, gene amplification and overexpression of YAP were shown to promote tumor progression. However, in ER-positive breast cancer, YAP protein was less expressed than normal breast tissue, and the level of YAP expression tended to be favorable for prognosis [32]. In contrast, recombinant viruses may induce higher apoptosis in absence of ER in MDA-MB-231 cells. Based on the above studies, we supposed that ER inhibited MCF-7 apoptosis induced by lectins through regulating signal pathways or antiviral immunity, but the specific mechanism needs to be further explored.

The replication mechanisms of recombinant viruses were in a cell type-dependent manner, and lectins have different effects in promoting viral replication. In our study, APL enhanced oncoVV replication depending on Raf/MEK/ERK signal pathway in MCF-7 cells, but the exact opposite was observed in MDA-MB-231 cells. JNK activity did not affect the vaccinia virus replication in MCF-7 cells while JNK inhibitors inhibited the replication of all recombinant viruses in MDA-MB-231 cells. In addition, oncoVV-AVL/APL replication in MDA-MB-231 cells depended on Hippo/YAP pathway, while the opposite phenomenon was observed in MCF-7 cells. PI3K inhibitor had no effect on all viruses replication in MCF-7 cells, but PI3K could significantly manipulate the replication of oncoVV-TTL/AVL/WCL in MDA-MB-231 cells. Many studies have shown that ER and PR can affect viral replication and antiviral response. For example, ER and PR can inhibit innate and acquired immunity, thus changing susceptibility to HIV [42]. ER have been found to inhibit Zika viruses infection and the replication process [43]. Similarly, our previous study found that AVL could repress the antiviral response in Huh7 cells to facilitate viral replication [20]. In summary, the replication mechanisms of recombinant viruses between MCF-7 and MDA-MB-231 cells were completely different, ER or PR may affect viral replication and antiviral immunity by changing the lectin-signaling pathways interaction in BC, and lectins have different effects in promoting viral replication, which may be related to ligand (lectins)-receptor interactions. Hovever, the specific mechanisms need to be further explored.’

Minor points:

Point 1: Introduction. When introducing the topic of oncolytic vaccinia virus (poxvirus), the authors cited reviews published in 2009; 2011 and 2015. In fact, there are much more recent comprehensive and authoritative reviews on the topic (published in 2019) that provide more up-to-date authoritative reviews and more suitable for citations for this purpose: (1). PMID: 30626434. (2). PMID: 30919708. Unlike some cited older papers, the authors of these recent review articles have been and continue working in the field for over 20 years and going and their reviews are more authoritative and up-to-date.

Response 1: Thanks for the suggestions. Some new reviews on the topic were added in the introduction section.

Point 2:. Methodology: Descriptions of some methodologies are not detailed enough. For example, when authors have used inhibitors (such as sorafenib) together with VV, when was the inhibitor added, and how often (just once)? A lot of detailed information need to be added.

Response 2: Thanks for kindly reminder. Some methodologies were rewrote, and more detailed information was added. The added sentences were highlighted with red color.

Point 3: Fig. 3 a,b. Based on the current plots, it seems that oncoVV does not replicate at all. This is very misleading. In addition, the description in the Results is also poor as there are no information on the basal level of replication (or accumulated virus) from oncoVV.

Response 3: The replication of oncoVV has improved as well, and we have redone Figure 3 to highlight the trend of oncoVV replication. In addition, we have added the description of the baseline in the Results, which was highlighted with red color.

Point 4: Figs 3-7. The authors stated that they conducted TCID50 assay to measure quantities of infectious virus, yet the Y-axis indicates pfu/ml. How did the authors convert to pfu? (It has not been mentioned at all). This should be described in Methods.

Response 4: Thanks for kindly reminder. Since the determination of plaque forming units is often poor in repeatability, TCID50 assay is used in more studies to calculate the number of virus infections. This method is based on the literature (DOI: 10.1016/s0166-0934(01)00316-0), and the conversion formula of the virus titer was 7 x 10(d+0.5) PFU/ml (d = sum of positive proportion). We have added this to the Materials and Methods section.

Point 5: Figs. 4-5. At which time point were the virus titers determined?

Response 5: We have added this to the Materials and Methods section, which was highlighted with red color.

Point 6: Fig. 6c. Lanes are not labeled. In addition, there should be 12 lanes, instead of 6 lanes.

Response 6: Thanks for kindly reminder. We have re-labeled the lanes. It was reported that the protein level of YAP was negatively correlated with the expression of ER/PR (PMCID: PMC9163075), and no obvious YAP bands were detected in MCF-7 cells.

Point 7: Discussion: Authors have discussed the confusing data collected from these two breast cancer cell lines (MCF7 and MDA-MB-231). However, one thing is clear that data drawn by the authors supported that statement that oncolytic virus combined with some specific inhibitor or activator of specific signaling pathway may work synergistically, leading to improved therapy. In this regard, there is a review to support your discussion (PMID: 36221123. Improving cancer immunotherapy by rationally combining oncolytic virus with modulators targeting key signaling pathways. Mol Cancer. 2022;21:196.).

Response 7: Thank you for your constructive comments. We have rewritten the discussion as your suggestion, and highlighted by red color. The details are as follows:

‘Key molecules in relevant signaling pathways had the potential to synergize with oncoVV-lectins to improve therapeutic efficacy. In MDA-MB-231 cells, we found that Sorafenib or U0126 significantly restrained oncoVV-AVL replication, and oncoVV-AVL infection triggered activation of cellular phosphorylation of ERK, suggesting that AVL improved viral replication by activating ERK pathway. Furthermore, the replication of oncoVV-AVL was dramatically reduced by SP600125 and KY12420, suggesting that JNK and PI3K could affect the replication of oncoVV-AVL as well. Besides, XMU-MP-1 significantly decreased the replication of oncoVV-AVL, and lower expression of YAP had been detected in MDA-MB-231 cells treated with oncoVV-AVL, suggesting that AVL enhanced replication by regulating the Hippo/YAP pathway. Based on these studies, we inferred that small-molecule drugs targeting key cellular signaling pathways, including MAPK, PI3K/Akt, and Hippo, could be conjunction with oncoVV-AVL to promote breast cancer therapy. In addition, key pathway factors of JNK and PI3K pathways may contribute to the efficacy of oncoVV-APL/TTL/WCL. However, the replication mechanism of recombinant viruses in MCF-7 cells was still poorly understood, and needed to be further explored.’

Point 8:  The following existing references need correct or additional information:

Ref #1. Ferlay J. et al. The correct page numbers are, 778-789.

Ref #18. Authors’ own paper. The correct volume number is 20. not 11.

Ref #25. Jiang Q. et al. The missing page numbers are, 17-28.

Response 7: We are sorry for these mistakes. We rechecked the format of all references and fixed some errors.

Round 2

Reviewer 2 Report

The authors have improved the quality of presentation and discussion significantly.